# Productivity of Three Pea (*Pisum sativum* L.) Varieties as Influenced by Nutrient Supply and Meteorological Conditions in Boreal Environmental Zone

**DOI:** 10.3390/plants12101938

**Published:** 2023-05-09

**Authors:** Daiva Janusauskaite

**Affiliations:** Department of Plant Nutrition and Agroecology, Institute of Agriculture, Lithuanian Research Centre for Agriculture and Forestry, Instituto al. 1, LT-58344 Akademija, Kėdainiai District, Lithuania; daiva.janusauskaite@lammc.lt

**Keywords:** NPK fertilization, seed yield, seed quality, semi-leafless pea, yield components

## Abstract

In order to grow crops that reduce the negative impact on the environment, as well as meet the nutritional needs of the increasing human population, it is necessary to include new and more sustainable production strategies into current agricultural systems. The aim of our study was to evaluate the optimal nutritional conditions of semi-leafless pea productivity and ascertain the influence of meteorological factors on the productivity of these plants under boreal environmental conditions. The test involved three semi-leafless pea varieties, one of which was a new variety, and eight N fertilization treatments were used: (1) without fertilizers (N_0_), (2) without N fertilizers (N_0_), (3) N_15_, (4) N_30_, (5) N_45_, (6) N_15+15_, (7) N_60_, and (8) N_60_. Plots of the second–seventh treatment received a base application of P_40_K_80_; the eighth treatment received P_80_K_160_. Fertilizer efficiency depended on the meteorological conditions. Based on their productivity, the pea varieties were arranged in the following descending order: Ieva DS ˃ Respect ˃ Simona. Compared with unfertilized peas, NPK fertilizers enhanced the seed yield by 10.6–12.9% on average. Splitting the N_30_ rate and applying N_60_, under a background of P_40_K_80_, was not efficient. The optimal rate of N_15–45_P_40_K_80_ fertilizers for peas was determined. Meteorological factors significantly influenced seed yield by 75.2%, 44.1%, and 79.9% for varieties Ieva DS, Simona, and Respect, respectively.

## 1. Introduction

Peas are cultivated in 84 countries around the world. The high prevalence of peas is due to their great yield potential, remarkable biological properties, and value for human and livestock nutrition. Pea seeds are high in nutritional value, rich in protein, carbohydrates, phosphorus, vitamins A and B, iron, and calcium, and are easily digested [1,2].

The goal of the “European Green Deal” is to identify ways to reduce the overabundance of nutrients in the environment that are a basic source of soil, water, and air pollution, and thus adversely affect biodiversity and climate. The target of the agricultural policy is to reduce fertilizer consumption by at least 20% by the year 2030 while ensuring retention of soil fertility [3]. One of the strategies to achieve these goals is the introduction of legumes, including peas, into crop rotation. The most essential profit of pea growing is the enrichment of soil with nitrogen collected during the nitrogen fixation process, which is also used by the succeeding crops [4,5]. Expanded pea cultivation would reduce the use of nitrogen from mineral fertilizers and maximize the use of biological nitrogen [6,7]. Residues of legumes contribute to the accumulation of organic matter in soil [8,9]. Like all legumes, peas activate the microbial functions of soil which are also related to the increase in organic C [8,10]. Aboveground legume biomass and belowground biomass directly contribute to soil organic carbon accumulation [10,11,12]. Peas are an important crop for a sustainable future and better food systems [6,13]. Peas are suitable as break crops in wheat-based rotations because they are not susceptible to the same diseases and pests as the cereal [11].

New pea types, the semi-leafless varieties, are characterized by improved disease tolerance, easier harvesting, and higher yield potential [14]. There is a lack of knowledge about the nutrient requirements of semi-leafless peas as they differ from leafy varieties. There are different opinions about pea fertilization. Little data are available on the nutrient requirements of the new semi-leafless cultivars and their response to NPK fertilization. Research is needed on the nutrient requirements of Lithuania’s newest pea cultivar Ieva DS and to compare the productivity with already cultivated varieties. The importance of nitrogen for pea is unquestionably high, as in all plants [4,15]. There is an opinion that it is enough to fertilize pea only with “starting” nitrogen rates [16]. However, other studies have shown that in soil with a high nitrate–nitrogen content, starter N application had a detrimental effect on pea germination, and nodulation and seed yield [17]. The efficiency of fertilizers is governed by meteorological conditions and the nutrient content in the soil [18,19]; therefore, recommended NPK rates can vary widely [20,21]. The phosphorus and potassium requirements of peas vary depending on the growing conditions [20]. Potassium stimulates the transport of nitrogen from root nodules to the aboveground parts and affects protein synthesis [22].

The response of different pea varieties to fertilization mostly varies [23,24,25].

When including peas in the crop sequence, seed inoculation with a specific Rhizobium strain is an important practice for optimizing N nutrition in legumes [1,5]. In previous studies, it was found that the inoculation of pea seed had a strong positive impact on pea productivity, especially in the fields where they were not previously cultivated [6]; however, it did not affect the yield of peas in the fields where they were previously grown [17]. We did not use inoculation because peas are included in our fields’ crop rotation.

This study aimed (I) to evaluate the productivity potential of semi-leafless peas and to determine the optimal NPK rates, and (II) to establish the relationship between pea productivity and its components and meteorological factors under boreal environmental conditions.

## 2. Results

### 2.1. Pea Seed Yield

Weather conditions differed in the amount of rainfall and its distribution between the experimental seasons (Table 1); therefore, the results were discussed separately for all experimental years. A two-way ANOVA showed that pea seed yield was influenced by variety (factor A) (*p* ≤ 0.01), fertilization (factor B) (*p* ≤ 0.01), and their interaction (A × B) (*p* ≤ 0.05 and *p* ≤ 0.01) (Table 1). There was one exception, when fertilization and A × B interaction did not have significant influence on seed yield in the 2015 year. This year was distinguished from the others by low rainfall and low HTC, which corresponded to the lowest limit of optimal irrigation. The lack of moisture negatively affected the pea yield and the efficiency of the fertilizers. 

Variety was the main factor determined, at 29.7–56.9% of the total variability of the seed yield. Fertilization was responsible for 18.3–34.0% of the differences between the treatments. The interaction A × B explained the least part of the differences (9.1–8.9%).

The variety Ieva DS showed the highest yield (3.88–5.16 t·ha^−1^), or 5.9–13.8% higher than the trial mean, regardless of the different weather conditions in the experimental years. The variety Respect exceeded the trial mean by 7.8% and 3.2% only under sufficiently wet weather conditions (HTC 1.6 at 2014 and 1.4 at 2017, respectively). The seed yield of the variety Simona was 9.3–13.8% lower than the trial mean, with one exception, when it exceeded the trial mean by 3.9% under the dry weather conditions in 2015.

The mean of the fertilized treatments varied from 3.65 to 4.98 t·ha^−1^. Depending on the year’s weather conditions, the efficiency of the fertilizers was not the same during different years. Under the effect of NPK fertilization, seed yield increased by 0.11–0.74 t·ha^−1^. 

A three-way ANOVA showed that the weather conditions of the experimental year (factor A) was the main factor responsible for 39.4% of the total variability of the averaged data of the seed yield (Table 2). Variety (factor B) explained 15.8%, fertilization (factor C) governed 11.2%, and A × B and A × C interactions explained 12.3% and 2.4%, respectively, of the yields’ averaged data variations. In all cases, the influence of the mentioned factors was significant at *p* ≤ 0.01. No significant effect of interactions B × C and A × B× C on seed yield was established.

The most productive pea variety was Ieva DS, which produced the average seed yield of 4.61 t·ha^−1^ and, compared with the control, exceeded the trial mean by 0.35 t·ha^−1^, or 8.2%. According to the average yield, varieties were arranged in the following descending order: Ieva DS ˃ Respect ˃ Simona.

NPK fertilizer increased the seed yield by 0.55 t·ha^−1^ (or 15.6%) on average, compared with unfertilized pea. The influence of fertilizers on seed yield was significant (*p* ≤ 0.05) in all cases. Under a background of P_40_K_80_, the application of N_15_, N_30_, and N_45_ rates increased seed yield by 0.42 t·ha^−1^, 0.57 t·ha^−1^, and 0.69 t·ha^−1^ (or 11.1%, 15.1%, and 18.3%, respectively), compared with the control (N_0_P_0_K_0_) (Figure 1). Splitting the N_30_ rate (N_15_ + N_15_) (sixth treatment) did not exceed its single application and yielded the same efficiency as N_15_ (third treatment). Seed yield did not increase with increasing the nitrogen fertilizer rate above N_45_. The most abundant fertilization (P_80_K_160_ + N_60_) resulted in the highest (4.57 t·ha^−1^) yield, or 20.9% higher, compared with unfertilized pea.

The seed yield of pea varieties was found to be significantly correlated with the N rate (Figure 2). The relationship was strong and ranged from 0.789 * to 0.956 *.

### 2.2. Seed Yield Components

The analysis of variance revealed that all seed yield components were significantly (*p* ≤ 0.01) influenced by year (factor A), variety (factor B), fertilization (factor C), and A × B interaction (Table 2). Year was the main factor determining 59.9% of the number of pods per plant (NPP), 57.0% of the number of seeds per plant (NPS), 29.1% of the weight of seeds per plant (WSP), and 62.1% of the number of pods per m^2^ (NPm^2^) total variability. During the wet year, 2014 (HTC = 1.6), the conditions were most favourable for the formation of seed yield components. Compared with the trial mean, NPP, NSP, and NPm^2^ were found to be significantly higher by 35.0%, 32.9%, and 38.0%, in 2014. Meanwhile, during other research years, yield components were less than the trial mean.

Variety explained 11.4%, 3.1%, 16.8%, and 4.3% of the total variability of NPP, NSP, WSP, and NPm^2^, respectively. The variety Ieva DS was distinguished from other varieties by its values of the yield components: NPP, WSP, and NPm^2^ were higher by 10.0%, 41.4%, and 7.7%, respectively, compared with the trial mean. Meanwhile NPP, NSP, WSP, and NPm^2^ for the variety Respect were lower by 13.3%, 5.6%, 24.3%, and 7.2% than the trial mean.

Fertilization resulted in the lowest part of the yield components variation and explained only 2.6%, 2.5%, 0.34%, and 3.9% of NPP, NSP, WSP, and NPm^2^, respectively, of their total variability. Fertilization resulted in 11.5%, 9.2%, 8.3%, and 10.1% higher NPP, NSP, WSP, and NPm^2^, respectively, compared with unfertilized peas.

### 2.3. Seed Quality

The thousand seed weight (TSW) was significantly (*p* ≤ 0.01) influenced by year (factor A), variety (factor B), fertilization (factor C), and A × B and A × C interactions (Table 3). Year was the major factor responsible for most (48.9%) of the TSW total variability. A × B interaction governed 36.2% of the differences between treatments. The influence of variety, fertilization, and A × C interaction was small and explained only 1.2–3.7% of the TSW variation. In 2014, the TSW was 9.5% lower than the trial mean; meanwhile, the TSW exceeded the trial mean by 4.3% and 4.7% in 2016 and 2017, respectively. NPK fertilization significantly decreased TSW by 2.7–3.5%. 

The seed protein content was significantly influenced (*p* ≤ 0.05 and *p* ≤ 0.01) by year (factor A), variety (factor B), fertilization (factor C), and A × B and A × C interactions (Table 3). The year explained a large part (44.7%) of the total variability and variety and A × B interactions were responsible for half as much variation (22.5% and 20.3%, respectively). Meanwhile, fertilization and A × C interactions determined only 2.2% and 1.1%, respectively, of the total variance of the protein content. Under the wettest conditions, in 2014, the protein content was 8.4% higher than the trial mean. The variety Simona significantly accumulated the highest amount of protein (by +4.0 percentage point (pp)), whereas the lowest amount of protein was accumulated by the variety Respect (by –4.4 pp), in comparison with the trial mean.

### 2.4. Root Nodules

The effect of year and A × B interaction on root nodules was significant (*p* ≤ 0.01) (Table 3). All other factors had no significant effect on this indicator. Under the normal moisture regime in 2016 and 2017, the number of root nodules was the highest, and exceeded the trial mean by 6.7% and 84.5%, respectively.

### 2.5. Relationship between Seed Yield, Yield Components, Seed Quality Indices, and Meteorological Indices

We ascertained a correlation between seed yield and yield components (Table 4). The data showed that seed yield for the variety Ieva DS significantly (*p* ≤ 0.05 and *p* ≤ 0.01) correlated with all yield components, the number of root nodules (NRN), the insertion height of the first pod (IFP), and seed quality indices. The correlation of seed yield for the variety Simona with the mentioned indices was not significant in all cases: NRN did not correlate with seed yield. No correlation was found between seed yield for the variety Respect and WSP, NRN, and protein content. Data, averaged across varieties, revealed that this correlation ranged from weak to moderate and was significant in most cases. 

A simple correlation analysis confirmed the existence of a correlation between seed yield, the morphological traits of peas, and meteorological indices throughout the growing season (Table 5). However, differences among the varieties were noticed in the strength of the relationship among the variables as shown by the values of the correlation coefficients. Averaged across varieties, seed yield positively corelated with precipitation (*p* ≤ 0.01) and HTC (*p* ≤ 0.01), and negatively correlated with sunshine duration (*p* ≤ 0.01). The accumulated growing degree days (AGDD) ˃ 5 °C, AGDD ˃ 10 °C, and HTC positively correlated with morphological traits and PC, except when AGDD ˃ 5 °C and AGDD ˃ 10 °C correlated negatively with NRN and TSW. Sunshine duration and relative air humidity negatively correlated with all indices in most cases.

The multiple linear regression model showed that seed yield, morphological traits, and quality indices were intensely influenced by weather conditions all through the growing season (Table 6). It was found that the interaction of meteorological factors influenced seed yield by 75.2%, 44.1%, and 79.9% for varieties Ieva DS, Simona, and Respect, respectively. Averaged across varieties, meteorological conditions caused from 50.6 to 87.4% of morphological traits and 76.2% root nodule data variation. Protein content and TSW were influenced by weather conditions by 48.6% and 52,6%, respectively.

## 3. Discussion

Fertilization is one of the basic ways to ameliorate the availability of soil nutrients to plants. The use of fertilizers is considered to be one of the most meaningful factors in enhancing crop yield. Fertilizing can positively change plant growth and productivity, but heavy uses of chemical fertilizers have created a variety of environmental and ecological problems. As a result, it is necessary to diversify crop rotation with legumes, such as peas, which are able to fix atmospheric nitrogen [6]. Their short growing period and ability to fix atmospheric nitrogen makes pea the best pre-crop for winter wheat. In addition, the plants not only ensure themselves on 2/3 of nitrogen, they also leave 60–100 kg of available nitrogen for subsequent culture [7]. There are different opinions about the fertilization of peas. Some researchers say that pea fertilizer rates depend on nutrients in the soil [20]. Huang et al. [17] found that when before sowing in soil and NO_3_-N was low (10 kg·ha^−1^), an application of starter N positively influenced pea yield (up to +19%); however, under higher initial soil NO_3_-N (44 kg·ha^−1^), the application of starter N reduced pea yield. Under no-tillage technology, N_25_P_30_K_40_ fertilization was recommended for peas [26]. The results of the present investigation revealed that the biggest part of the variation in yield data was determined by the year’s meteorological conditions. Weather conditions of the experimental year influenced the effectiveness of fertilizers. According to average data, the application of N_15_–N_45_ rates increased seed yield by 11.1–18.3% under a background of P_40_K_80_, compared with unfertilized peas. After increasing the N rate to N_60_, the seed yield did not increase. The highest N_60_P_80_K_160_ rate increased the yield, but the difference was not significant, compared with N_60_ under a background of P_40_K_80_. The N_15_–N_45_ rate was found to be economically and ecologically optimal for peas, grown under boreal conditions, which is close to the above-mentioned studies. There are conflicting data showing that doses of N from mineral fertilizers can vary from 73 to 97 kg·ha^−1^ N [27].

Abundant N fertilization has a negative effect on root nodule formation [17], which can affect subsequent nitrogen fixation [6]. Higher N rates reduce pea germination [17]. There are studies that show that germinating peas can only tolerate the 10 kg N ha^−1^ rates [16]. We found that fertilization had no significant effect on root nodules. This index was significantly influenced only by the weather conditions in the experimental year. The most favourable conditions for the formation of root nodules were at HTC 1.0–1.3, which means optimal irrigation. According to Khan et al. [23], the number of root nodules significantly differed between four tested varieties. In fact, in contrast with him, we found that the three tested pea varieties had no significant differences in the number of root nodules.

Pea varieties respond differently to fertilization [23,28] and environmental conditions [2]. The main traits that determine the level of adaptability in peas are a high harvest index, good ripening, disease resistance, resistance to shatter, high potential yield, seed weight, and seed number per pod [25,29]. In this study, the variety Ieva DS demonstrated the greatest adaptability and the ability to produce the highest seed yield of all studied varieties, regardless of the weather conditions. The highest values of yield components, such as NPP, WSP, and NPm, were also established for the variety Ieva DS. The highest TSW and protein content was for varieties Ieva DS and Simona. 

The growth, development, and productivity of plants are strongly affected by weather conditions [30]. In a study with eight pea cultivars, it was found that seed yield had a strong relationship with weather factors and the reproductive phase was limited by stress [31]. The most important factor that determined seed yield and protein content was the sum of the rainfall over the vegetation period [32]. Kuznetsov et al. [30] established a moderate relationship (r = 0.486) between pea yield and the amount of precipitation. Grabowska and Banaszkiewicz [33] found that the influence of air temperature and atmospheric precipitations on the yield of different varieties differed and caused from 82% to 87% of seed yield variability. The results of the current experiment revealed that the most favourable conditions for the seed yield components were in the wet years. Correlation analysis among the yield components and meteorological indices revealed that HTC had the greatest positive influence on the seed yield for the variety Respect. HTC influenced seed yield by 57.1%, 43.4%, and 23.4% for varieties Respect, Ieva DS, and Simona, respectively. Our findings are in line with previous results [33], stating that water stress is the key cause of a decrease in pea yield in a temperate climate. Other studies have also found a positive correlation between pea yield and HTC [30]. 

We found that the relationship between sunshine duration and seed yield was negative and varied from weak to strong. Sunshine duration determined 12.9%, 51.1%, and 52.0% of seed yield data variation, respectively, for varieties Simona, Ieva DS, and Simona. Other researchers found that, on the contrary, the lowest average sunshine duration related to a lower seed yield for faba beans [34].

The results obtained in this study show that genotype had a significant influence on the levels of protein content in pea seeds. The variety Simona was distinguished from the others with the highest protein content, the protein content was very similar for the variety Ieva DS, and the variety Respect significantly accumulated the lowest content of protein. This confirms previous findings in the literature that differences in climate, soil, varieties, and agronomic practices may cause a different yield and protein content [22,32,35]. We found that fertilizers did not have a significant influence on protein content. This is in good agreement with previous findings in the literature [18]. 

## 4. Materials and Methods

### 4.1. Site and Soil Description

A field experiment was carried out during 2014–2017 at the Institute of Agriculture, Lithuanian Research Centre for Agriculture and Forestry in Central Lithuania (55°23′50″ N and 23°51′40″ E). The locality is situated in a boreal environmental zone, where the average annual air temperature is 6.4 °C and the long-term annual precipitation is 572 mm. The soil of the experimental site was Endocalcari-Epihypogleyic Cambisol. The mean soil characteristics (at 0–25 cm sampling depth), determined annually before sowing, were as follows: pH_KCl_ 5.4–7.2 (potentiometrically), available phosphorus 84–150 mg kg^−1^ (A-L method), available potassium 140–186 mg kg^−1^ (A-L method), humus 1.5–2.2% (Tyurin method). The content of mineral nitrogen was 43–59 kg·ha^−1^ in 0–40 cm soil layer (as sum of N-NO_3_ and N-NH_4_, N-NO_3_—ionometrically, N-NH_4_—spectrophotometrically.) 

### 4.2. Experimental Details and Agronomic Management

The experiment involved three semi-leafless pea (*Pisum sativum* L.) varieties: Ieva DS, Simona, and Respect. Eight N fertilization treatments were used: (1) without fertilizers (N_0_), (2) without N fertilizers (N_0_), (3) 15 kg N·ha^−1^ (N_15_), (4) 30 kg N·ha^−1^ (N_30_), (5) 45 kg N·ha^−1^ (N_45_), (6) 15 + 15 kg N·ha^−1^ (N_15+15_), (7) 60 kg N·ha^−1^ (N_60_), (8) 60 kg N·ha^−1^ (N_60_). Nitrogen, as ammonium nitrate (NH_4_NO_3_) (34% N), was applied pre-sowing, with the exception of the 6th treatment, where additional 15 kg N·ha^−1^ was applied at stem elongation stage (BBCH 30–49). In conjunction with N fertilizer application, plots of 2nd–7th treatments received a base application of phosphate (P) fertilizer and potassium (K) fertilizer at 40 kg P·ha^−1^ and 80 kg K·ha^−1^, and plots of 8 treatment received double rate of PK, i.e., 80 kg P·ha^−1^ and 160 kg K·ha^−1^. P fertilizers were applied as granular superphosphate (Ca(H_2_PO_4_)_2_H_2_O) with P_2_O_5_ concentration of 20% and K as potassium chloride (KCl), with K_2_O concentration of 60%. All fertilizers applied before sowing were incorporated into the soil. 

The field trial was conducted using a split plot design with four replications, in which pea varieties were laid out in the main plot and the fertilization treatments in the subplot. The plot size was 15.0 m^2^ (1.5 m × 10.0 m).

Spring barley (*Hordeum vulgare* L.) was a pre-crop in all experimental years. Peas were sown on 15 April in 2014 and 2015, 11 April in 2016, and 18 April in 2017 at a density of 1.2 million viable seeds per hectare with a 13 cm distance between rows and an 8 cm distance in the rows. After sowing, herbicide Fenix (a.i. aclonifen, 600 g L^−1^) (3.0 L·ha^−1^) was sprayed each year. Chemical insect control was performed using the following products at BBCH 14–16: Fastac 50 (a.i. alfa-cipermetrin 50 g L^−1^) (0.20 L·ha^−1^) in 2014 and 2015 and Decis Mega 50 EW (i.e., deltametrin 50 g L^−1^) (0.15 L·ha^−1^) in 2016 and 2017.

### 4.3. Plant Sampling and Measurements

The number of pods per plant (NPP), number of seeds per plant (NSP), and weight of seeds per plant (WSP) were established from randomly selected ten plants of each plot. The number of pulses per m^2^ (NPm) was determined on plant samples collected from 100 cm long sections of two rows (0.50 m^2^) from each plot. All indices were assessed at pea physiological maturity. Number of root nodules (NRN) per plant was determined from randomly selected five plants of each plot at inflorescence emergence (BBCH 51–59).

### 4.4. Seed Yield and Seed Quality Analyses

Each year, the plots were harvested within the first ten-day period of August at complete maturity (BBCH 89) using a plot harvester “Wintersteiger Delta” (Ried im Innkreis, Germany). Seed yield (SY) of pea was adjusted to a 15% moisture content. Protein content (PC) from each plot, from samples selected after harvesting, were measured using an Infratec 1241 grain analyzer (Foss, Hilleroed, Denmark). Thousand seed weight (TSW) was counted using a seed counter “Contador” (Pfeuffer GmbH, Kitzingen, Germany) from four samples of 250 seeds per each plot.

### 4.5. Meteorological Conditions

The locality is situated in boreal climatic zone, with an average annual air temperature of 6.4 °C and a long-term annual precipitation of 568 mm. Rainfall and mean air temperature at the experimental site over the two growing seasons are provided in Figure 3. The conditions of the plant growing season were described using the hydrothermal coefficient (HTC) as the agrometeorological indicator, which was calculated according to the formula [36]:HTC = Σ p/0.1 Σ t, (1)
where Σ p represents the sum of precipitation (mm) during the test period, when the average daily air temperature is above 10 °C, and Σ t denotes the sum of active temperatures (°C) during the same period. If HTC > 1.6, the irrigation is excessive; HTC = 1.0–1.5 optimal irrigation; HTC = 0.9–0.8 weak drought; HTC = 0.7–0.6 moderate drought (arid); HTC = 0.5–0.4 heavy drought; and HTC < 0.4 very heavy drought [36].

Rainfall differed between the growing seasons, and the amount of rainfall totaled 212 mm (HTC = 1.6), 134 mm (HTC = 1.0), 215 mm (HTC = 1.3), 229 mm 212 mm (HTC = 1.4) in 2014, 2015, 2016 and 2017, respectively.

### 4.6. Statistical Analysis

A three-way ANOVA was used to determine the effects of year, variety, and fertilization treatment, on the morphological traits, SY, and seed quality parameters. The data were compared using Fisher’s least significant difference (LSD) test at the probability levels *p* ≤ 0.05 and *p* ≤ 0.01. Standard statistical procedures were used for calculating simple correlation coefficients. The statistical analysis was performed using Stat Eng from the statistical data processing package Selekcija.

## 5. Conclusions

It was found that the experimental year determined 39.4%, variety—15.8%, and fertilization—11.2%, of the yield averaged data variations. The influence of fertilizers was more strongly expressed in wet conditions compared with dry conditions. The lack of moisture negatively influenced the pea seed yield. The new pea variety, Ieva DS, showed the highest seed yield of all varieties, compared with the trial mean, and is suitable for cultivation in the boreal environmental zone. According to seed yield, the varieties were arranged in the following descending order: Ieva DS ˃ Respect ˃ Simona. NPK fertilizers increased the seed yield by 0.40–0.78 t·ha^−1^ (or 10.6–12.9%) on average, compared with unfertilized peas. Under a background of P_40_K_80_, the application of N_60_ and splitting the N_30_ rate into N_15+15_ was not efficient. The optimal rate of _N15–45_P_40_K_80_ fertilizer was determined when peas were grown in soil containing moderate amounts of available phosphorus and available potassium. The meteorological factors had a significant impact on yield components and pea seed quality, and significantly influenced the seed yield by 75.2%, 44.1%, and 79.9% for varieties Ieva DS, Simona, and Respect, respectively. 

## Figures and Tables

**Figure 1 plants-12-01938-f001:**
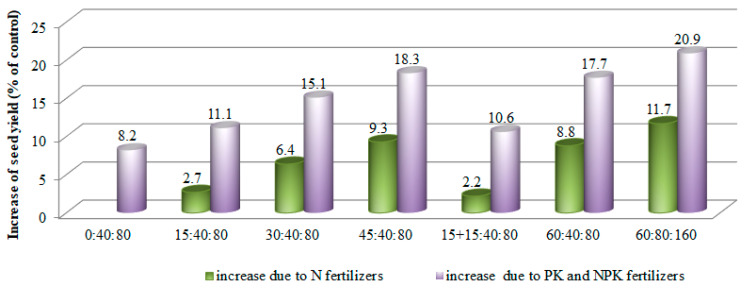
The influence of fertilization on increase in seed yield (% of control) (averaged across varieties).

**Figure 2 plants-12-01938-f002:**
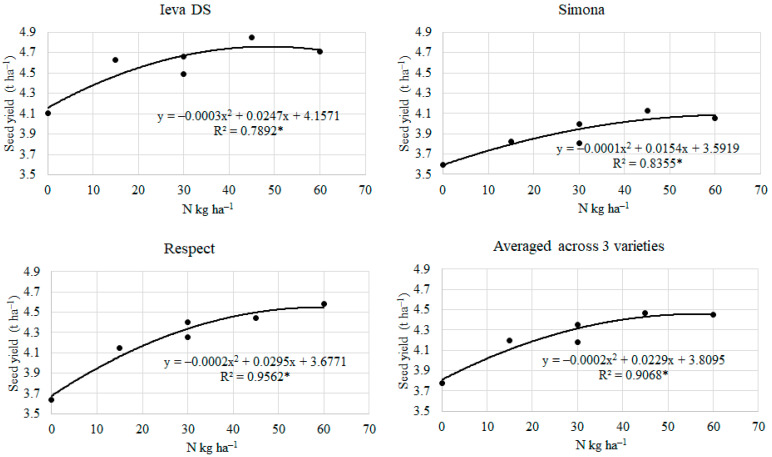
The relationship between N rate and pea seed yield (averaged 2014–2017). * significant at *p* ≤ 0.05.

**Figure 3 plants-12-01938-f003:**
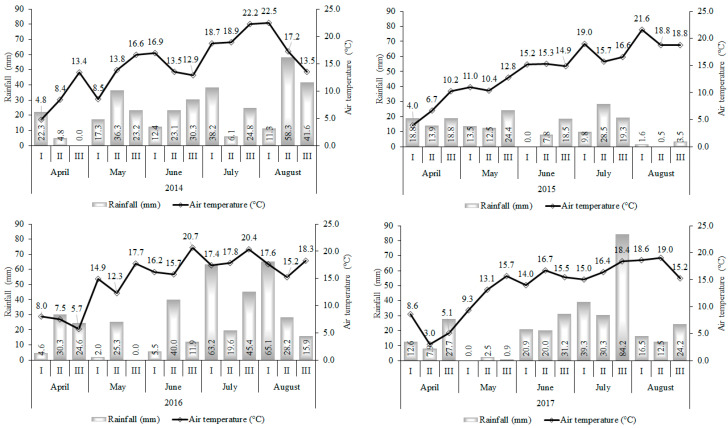
Rainfall and temperature distribution during the growing season.

**Table 1 plants-12-01938-t001:** Influence of variety and fertilization on seed yield (t·ha^−1^) of pea.

Variety (Factor A)	Fertilization (Factor B)	2014	2015	2016	2017
The effect of variety					
Ieva DS		5.16 a^§^	3.88 a^§^	4.70 a^§^	4.69 a^§^
Simona		4.19 c	3.77 a	3.68 c	4.02 c
Respect		5.24 a	3.25 c	4.02 c	4.57 a
Trial mean		4.86	3.63	4.13	4.43
The effect of fertilization					
	1. NPK 0:0:0	4.29 b^†^	3.53 b^†^	3.58 b^†^	3.78 b^†^
	2. 0:40:80	4.72 a	3.44 b	3.85 a	4.35 a
	3. 15:40:80	4.88 a	3.61 b	3.96 a	4.37 a
	4. 30:40:80	4.95 a	3.64 b	4.16 a	4.66 a
	5. 45:40:80	5.10 a	3.75 b	4.40 a	4.64 a
	6. 15 + 15:40:80	4.76 a	3.60 b	4.10 a	4.27 a
	7. 60:40:80	4.99 a	3.73 b	4.39 a	4.68 a
	8. 60:80:160	5.24 a	3.75 b	4.62 a	4.67 a
Mean of NPK fertilized treatments		4.98	3.65	4.21	4.52
Differences fertilized NPK (3–8) vs. unfertilized (1)		0.69	0.11	0.63	0.74
Contribution (%) of sum squares) of variety (factor A), fertilization (factor B), and their interaction					
A		56.9 **	29.7 **	56.2 **	34.0 **
B		18.3 **	4.3	30.8 **	34.0 **
A × B		8.9 *	1.8	6.1 **	8.5 *

Different letters in column denote a statistically significant difference (at *p* ≤ 0.05 according to LSD): ^§^—between treatments and trial mean; ^†^—between treatments; * *p* ≤ 0.05; ** *p* ≤ 0.01.

**Table 2 plants-12-01938-t002:** Influence of year, variety, and fertilization on seed yield and yield components of pea (averaged 2014–2017).

Year (Factor A)	Variety (Factor B)	Fertilization (Factor C)	Number of Pods Per Plant	Number of Seeds Per Plant	Weight of Seeds Per Plant (g)	Number of Pods Per m^2^	Seed Yield (t·ha^−1^)
The effect of years							
2014			8.1 a^§^	29.9 a^§^	6.4 c^§^	839 a^§^	4.86 a^§^
2015			4.9 c	19.6 c	5.3 c	450 c	3.63 c
2016			5.1 c	18.6 c	4.9 c	568 c	4.13 c
2017			5.7 c	22.1 b	11.6 a	574 c	4.43 a
The effect of variety							
	Ieva DS		6.6 a^§^	22.6 b^§^	9.9 a^§^	655 a^§^	4.61 a^§^
	Simona		6.0 b	23.8 a	5.9 c	604 b	3.91 c
	Respect		5.2 c	21.2 c	5.3 c	564 c	4.27 b
Trial mean			6.0	22.5	7.0	608	4.26
The effect of fertilization							
		1. NPK 0:0:0	5.4 b^†^	20.7 b^†^	6.5 b^†^	552 b^†^	3.78 b^†^
		2. 0:40:80	5.7 b	21.7 b	6.7 b	562 b	4.09 a
		3. 15:40:80	5.8 a	22.1 a	6.9 b	593 a	4.20 a
		4. 30:40:80	6.1 a	22.5 a	7.3 a	607 a	4.35 a
		5. 45:40:80	6.1 a	23.1 a	7.3 a	614 a	4.47 a
		6. 15 + 15:40:80	6.2 a	23.3 a	7.3 a	629 a	4.18 a
		7. 60:40:80	6.1 a	23.0 a	7.1 a	644 a	4.45 a
		8. 60:80:160	6.2 a	23.8 a	7.4 a	662 a	4.57 a
Mean of NPK fertilized treatments (3–8)			6.1	23,0	7.2	625	4.37
Differences fertilized NPK (3–8) vs. unfertilized (1)			0.7	2.1	0.7	73	0.55
Contribution (%) of sum squares) of year (factor A), variety (factor B), fertilization (factor C), and their interaction							
A			59.9 **	57.0 **	29.1 **	62.1 **	39.4 **
B			11.4 **	3.1 **	16.8 **	4.3 **	15.8 **
C			2.6 **	2.5 **	0.34 **	3.9 **	11.2 **
A × B			7.2 **	13.3 **	48.7 **	4.5 **	12.3 **
A × C			0.8	1.6	0.4	3.8 **	2.4 **
B × C			1.0	1.1	0.2	1.1	1.0
A × B × C			1.7	2.4	0.7	1.9	2.25

Different letters in column denote a statistically significant difference (at *p* ≤ 0.05 according to LSD): ^§^—between treatments and trial mean; ^†^—between treatments; ** *p* ≤ 0.01.

**Table 3 plants-12-01938-t003:** Influence of year, variety, and fertilization on thousand seed weight (TSW), protein content of pea, and root nodules per plant (averaged 2014–2017).

Year (Factor A)	Variety (Factor B)	Fertilization (Factor C)	TSW (g)	Protein (%)	Root Nodules (Number Per Plant)
The effect of years					
2014			229 c^§^	24.4 a^§^	8.4 b^§^
2015			253 b	21.7 c	17.0 a
2016			264 a	21.8 c	24.8 a
2017			265 a	22.0 c	42.8 a
The effect of variety					
	Ieva DS		258 a^§^	22.7 a^§^	23.9 b^§^
	Simona		248 a	23.4 a	24.6 b
	Respect		252 c	21.5 c	21.2 b
	Trial mean		253	22.5	23.2
The effect of fertilization					
		1. NPK 0:0:0	259 b^†^	23.0 b^†^	25.4 b^†^
		2. 0:40:80	252 c	22.1 c	24.9 b
		3. 15:40:80	251 c	22.4 c	24.4 b
		4. 30:40:80	252 c	22.5 c	23.7 b
		5. 45:40:80	253 c	22.5 c	20.8 b
		6. 15 + 15:40:80	251 c	22.6 c	23.2 b
		7. 60:40:80	253 c	22.5 c	20.4 b
		8. 60:80:160	250 c	22.3 c	23.1 b
Mean of NPK fertilized treatments (3–8)			252	22.4	22.6
Differences fertilized NPK (3–8) vs. unfertilized (1)			−8	−0.6	−2.8
Contribution (%) of sum squares) of year (A), variety (B), fertilization (C), and their interaction					
A			48.9 **	44.7 **	50.7 **
B			3.7 **	22.5 **	0.7
C			1.8 **	2.2 **	0.9
A × B			36.2 **	20.3 **	4.4 **
A ×C			1.2 **	1.1 *	1.9
B × C			0.5	0.5	2.7
A × B × C			0.7	0.8	5.3

Different letters in column denote a statistically significant difference (at *p* ≤ 0.05 according to LSD): ^§^—between treatments and trial mean; ^†^—between treatments; * *p* ≤ 0.05; ** *p* ≤ 0.01.

**Table 4 plants-12-01938-t004:** Correlation analysis among seed yield, yield components, and seed quality indices.

Index	Ieva DS	Simona	Respect	Mean across Varieties
PH	0.726 **	0.663 **	0.759 **	0.598 **
NPP	0.838 **	0.699 **	0.672 **	0.570 **
NSP	0.767 **	0.712 **	0.548 **	0.423 **
WSP	0.417 *	0.683 **	0.560	0.380 **
NPm	0.840 **	0.629 **	0.880 **	0.696 **
IFP	0.592 **	0.660 **	0.802 **	0.620 **
NRN	−0.369 *	−0.116	−0.035	−0.167
PC	0.690 **	0.509 **	−0.120	0.138
TSW	−0.813 **	−0.385 **	−0.529 **	−0.437 **

PH—plant height, NPP—number of pods per plant, NSP—number of seeds per plant, WSP—weight of seeds per plant, NPm—number of pods per m^2^, IFP—insertion height of the first pod, NRN—number of root nodules, PC—protein content, TSW—thousand seed weight; * *p* ≤ 0.05; ** *p* ≤ 0.01.

**Table 5 plants-12-01938-t005:** Correlation analysis among the yield components and meteorological indices.

Variety	Indices	Precipitation	AGDD ˃ 5 °C	AGDD ˃ 10 °C	Sunshine Duration	Relative Air Humidity	HTC
Ieva DS	SY	0.501 **	0.290	0.200	−0.715 **	0.103	0.659 **
	PH	−0.007	0.357 *	0.426 *	−0.480 **	−0.455 **	0.600 **
	NPP	0.127	0.501 **	0.502 **	−0.496 **	−0.166	0.525 **
	NSP	0.228	0.264	0.297	−0.533 **	−0.332	0.615 **
	WSP	0.012	−0.241	−0.151	−0.332	−0.530 **	0.486 **
	NPm	0.328	0.365 *	0.320	−0.615 **	−0.033	0.601 **
	IFP	−0.086	0.093	0.214	−0.437 *	−0.671 **	0.627 **
	NRN	0.334	−0.857 **	−0.863 **	−0.230	−0.414	0.278
	PC	0.118	0.392 *	0.421 *	−0.555 **	−0.313	0.628 **
	TSW	−0.145	−0.709 **	−0.657 **	0.387 *	−0.113	−0.331
Simona	SY	0.004	0.020	0.093	−0.359 *	−0.448 *	0.484 **
	PH	−0.251	0.326	0.460 **	−0.280	−0.641 **	0.462 **
	NPP	−0.026	0.324	0.401 *	−0.466 **	−0.486 **	0.595 **
	NSP	−0.033	0.261	0.349	−0.473 **	−0.539 **	0.619 **
	WSP	0.223	−0.199	−0.134	−0.653 **	−0.553 **	0.804 **
	NPm	−0.054	0.562 **	0.605 **	−0.364 *	−0.284	0.432 *
	IFP	−0.507 **	0.300	0.487 **	−0.007	−0.753 **	0.232
	NRN	0.614 **	−0.776 **	−0.857 **	−0.472 **	0.079	0.444 **
	PC	−0.041	0.476 **	0.535 **	−0.422 *	−0.379 *	0.519 **
	TSW	0.839 **	−0.116	−0.366 *	−0.449 **	0.838 **	0.182
Respect	SY	0.373 *	0.171	0.149	−0.721 **	−0.190	0.755 **
	PH	−0.211	0.469 **	0.571 **	−0.281	−0.499 **	0.419 *
	NPP	−0.245	0.639 **	0.716 **	−0.176	−0.338	0.265
	NSP	−0.438 *	0.608 **	0.729 **	0.016	−0.445 *	0.112
	WSP	−0.404 *	0.370 *	0.516 **	−0.056	−0.597 **	0.231
	NPm	0.233	0.494 **	0.465 **	−0.566 **	−0.074	0.565 **
	IFP	−0.122	0.245	0.357 *	−0.401 *	−0.610 **	0.571 **
	NRN	0.747 **	−0.765 **	−0.884 **	−0.564 **	0.200	0.496 **
	PC	−0.592 **	0.717 **	0.791 **	0.487 **	−0.051	−0.466 **
	TSW	0.300	−0.815 **	−0.867 **	0.038	0.166	−0.077
Mean	SY	0.292 **	0.153	0.131	−0.550 **	−0.121	0.568 **
across	PH	−0.150	0.360 **	0.458 **	−0.331 **	−0.512 **	0.472 **
varieties	NPP	−0.026	0.410 **	0.458 **	−0.367 **	−0.314 **	0.446 **
	NSP	−0.096	0.330 **	0.408 **	−0.331 **	−0.431 **	0.448 **
	WSP	−0.037	−0.018	0.074	−0.349 **	−0.526 **	0.499 **
	NPm	0.141	0.467 **	0.462 **	−0.482 **	0.140	0.504 **
	IFP	−0.234 *	0.209 *	0.348 **	−0.280 **	−0.670 **	0.473 **
	NRN	0.574 **	−0.783 **	−0.855 **	−0.429 **	0.061	0.408 **
	PC	−0.076	0.401 **	0.443 **	−0.239 *	−0.241 *	0.300 **
	TSW	0.366 *	−0.494 **	−0.587 **	−0.056	0.322 **	−0.037

AGDD—accumulated growing degree days, HTC—hydrothermal coefficient; SY—seed yield; PH—plant height, NPP—number of pods per plant, NSP—number of seeds per plant, WSP—weight of seeds per plant, NPm—number of pods per m^2^, IFP—insertion height of the first pod, NRN—number of root nodules, PC—protein content, TSW—thousand seed weight; * *p* ≤ 0.05; ** *p* ≤ 0.01.

**Table 6 plants-12-01938-t006:** Correlation coefficient (R) of the multiple correlation between pea seed yield, morphological traits, quality indices, and meteorological indices.

Parameters (y)	Ieva DS		Simona		Respect		Mean across Varieties	
	R	F_fact._	R	F_fact._	R	F_fact._	R	F_fact._
SY	0.867	12.63 **	0.664	3.29 **	0.894	16.50 **	0.684	13.03 **
PH	0.982	115.25 **	0.991	216.48 **	0.983	119.22 **	0.931	97.14 **
NPP	0.928	25.65 **	0.971	67.87 **	0.965	56.53 **	0.841	35.88 **
NSP	0.864	12.32 **	0.973	72.88 **	0.943	33.37 **	0.839	35.28 **
WSP	0.648	30.20 *	0.906	18.97 **	0.886	125.27 **	0.711	15.18 **
NPm	0.857	11.54 **	0.958	46.80 **	0.935	28.72 **	0.873	47.36 **
IFP	0.959	47.68 **	0.974	76.07 **	0.974	77.88 **	0.935	103.14 **
NRN	0.869	12.88 **	0.885	15.00 **	0.954	42.07 **	0.873	47.30 **
PC	0.968	62.10 **	0.985	131.61 **	0.826	8.91 **	0.697	14.01 **
TSW	0.937	30.16 **	0.976	82.84 **	0.985	138.59 **	0.725	16.41 **

Multi regression equation y = a + bx^1^ + cx^2^ + dx^3^ + ex^4^ + fx^5^, where y—SY, or morphological parameters, or PC, or TSW, x^1^—precipitation, x^2^—AGDD—accumulated growing degree days ˃ 5 °C, x^3^—AGDD—accumulated growing degree days ˃ 10 °C, x^4^—sunshine duration, x^5^—relative air humidity. SY—seed yield, PH—plant height, NPP—number of pods per plant, NSP—number of seeds per plant, WSP—weight of seeds per plant, NPm—number of pods per m^2^, IFP—insertion height of the first pod, NRN—number of root nodules, PC—protein content, TSW—thousand seed weight. *, **—relationship between indices is significant at *p* ≤ 0.05 and *p* ≤ 0.01, respectively.

## Data Availability

Not applicable.

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
