# Peer review of "Productivity of Three Pea (*Pisum sativum* L.) Varieties as Influenced by Nutrient Supply and Meteorological Conditions in Boreal Environmental Zone"

_plants, 2023, doi:10.3390/plants12101938_

Round 1

Reviewer 1 Report

This paper aimed to understand the productivity of three pea (Pisum sativum L.) varieties as influenced by nutrient management practices and correlation with meteorological conditions in boreal environmental zone.

The title is inappropriate as it displayed the significance of this study, but not the novel points or main conclusions. I suggest to the authors to change the title to be more specific, explanatory and simplified in order to be easily understandable from the readers.

English should be improved throughout the manuscript.

Quantitative information of the results should be provided in the abstract.

The novelty is not sufficiently explained in the introduction section. Also, the researches gap should be clearly described.

Lines 31-35, 60-63 need to be rephrased.

 How many biological and technical replicates have you used? How many plants or amount of samples were taken for the measurement of each parameter? This is surely very important.

The quality of the discussion section must be improved. The authors should organize the discussion from the general to the specific, linking your findings to the literature, then to theory, then practice and practical application and avoid repetition of the sentences. Please mention clearly how your work is important and different in comparison to already reported work in this area.

Conclusions are not well supported by the study results. Future prospect statement and limitations of the study must be added in the conclusion section.

Overall, the authors did substantial work and got lots of data in this work. I think the current manuscript should be carefully and seriously revised to meet the standard for publication in Plants.

Extensive editing of English language is required

Author Response

A list of amendments to the Manuscript ID: plants- 2369985

Daiva Janusauskaite

Productivity of three pea (Pisum sativum L.) varieties as influenced by nutrient management

practices and correlation with meteorological conditions in boreal environmental zone

Response to Reviewer 1

Dear colleagues,

we are very appreciative to you for your valuable remarks, advice and your time spared for my manuscript. We have taken the comments into consideration and have made amendments accordingly.

Please find below our detailed description of the corrections.

Point 1. The title is inappropriate as it displayed the significance of this study, but not the novel points or main conclusions. I suggest to the authors to change the title to be more specific, explanatory and simplified in order to be easily understandable from the readers.

Response 1. We changed the title of manuscript.

Point 2. English should be improved throughout the manuscript.

Response 2. English was improved throughout the manuscript.

Point 3. Quantitative information of the results should be provided in the abstract.

Response 3. The main quantitative indicators are already presented in the summary: “In comparison with unfertilized pea, NPK fertilizers enhanced the seed yield by 10.6 ‒ 12.9% on average.” And “The meteorological factors significantly influenced seed yield by 75.2%, 44.1% and 79.9% for varieties Ieva DS, Simona, and Respect, respectively.” The quantitative values of the indicators are presented in the Results section and the conclusions, and this is presented very concisely in the summary. I think the summary match the requirements. In addition, the abstract is limited to 200 words. Therefore, listing more results here will exceed the word limit.

Point 4. The novelty is not sufficiently explained in the introduction section. Also, the researches gap should be clearly described.

Response 4. We added new information about researches gap.

Point 5. Lines 31-35, 60-63 need to be rephrased.

Response 5. I don't really understand what kind of rephrasing the reviewer wants.

                     We corrected L 31-35 and L 60-63.

Point 6.  How many biological and technical replicates have you used? How many plants or amount of samples were taken for the measurement of each parameter? This is surely very important.

Response 6. The field experiment was performed in 4 replicates. Everything is described in section 4. Materials and Methods :

“4.3. Plant sampling and measurements

The number of pulses per plant (NPP), number of seed per plant (NSP) and mass of seed per plant (MSP) were established from randomly selected ten plants of each plot. The number of pulses per m2 (NPm) was determined on plant samples collected from 100 cm long sections of two rows (0.50 m2) from each plot. All indices were assessed at pea physi-ological maturity. Number of root nodules (NRN) per plant was determined from random-ly selected five plants of each plot at inflorescence emergence (BBCH 51-59).

4.4. Seed yield and seed quality analyses

Each year, the plots were harvested within the first ten-day period of August at com-plete maturity (BBCH 89) with a plot harvester “Wintersteiger Delta” (Germany). Seed yield (SY) of pea was adjusted to a 15% moisture content. Protein content (PC) from each plot, from the samples selected after harvesting, were measured using the grain analyzer Infratec 1241 (Foss, Hilleroed, Denmark). A thousand seed weight (TSW) was counted with a seed counter “Contador” (Pfeuffer GmbH, Kitzingen, Germany) from four samples of 250 seeds per each plot.”

If it says "from each plot", it means in 4 replications.

Point 7. The quality of the discussion section must be improved. The authors should organize the discussion from the general to the specific, linking your findings to the literature, then to theory, then practice and practical application and avoid repetition of the sentences. Please mention clearly how your work is important and different in comparison to already reported work in this area.

Response 7. The opinion of the second reviewer was the opposite - "The discussion is quite detailed. This chapter clarifies (very well) some details not so well clarified in the results". If the author has the right to his opinion, I leave the Discussion chapter as it is now.

Point 8. Conclusions are not well supported by the study results. Future prospect statement and limitations of the study must be added in the conclusion section.

Response 8. In conclusions, main significant results of the study are presented in a clear and concise manner. I don't understand why the reviewer demands in the conclusions: "Future prospect statement and and limitations of the study must be added in the conclusion section". The studio answered the tasks and it can be seen in the conclusions. Future prospect statement is not a necessary part of the conclusions. Therefore, I corrected the conclusions minimally.

Point 9. Overall, the authors did substantial work and got lots of data in this work. I think the current manuscript should be carefully and seriously revised to meet the standard for publication in Plants.

Response 9. We revised the manuscript based on the reviewer's comments. We disagreed with some of the comments, which we will explain in the answers above.

Comments on the Quality of English Language

Extensive editing of English language is required

Response. We correct the mistakes in the English language.

Reviewer 2 Report

REVIEW REPORT PLANTS-2369985

TITLE:
Productivity of three pea (Pisum sativum L.) varieties as influenced by nutrient management
practices and correlation with meteorological conditions in boreal environmental zone

AUTHORS:
Daiva Janusauskaite

Reviewer: ID

Comments and Suggestions for Authors (19/04/2023)

1. Abstract:
Very good and clear
Referred to all topics of the work.
Very well explains the research carried out
It should be mentioned where trials were carried out (experiments) and the number of years
The keywords are very well chosen.

1. Introduction:
Good presentation and although not very extensive it is enough to clarifying and understand
the rest of the paper.
Missing square parenthesis on line 44 = [6.13]
It presents 25 references that are very connected to the theme and with a sense of
opportunity.

2. Results:
The results meet the expectations and objectives of the program and are very complete;
however, there were some comments:
1. They are very long and difficult to interpret and read, but after several readings it
becomes understandable.
2. Figure 1 (line 71) does not agree with the text
3. The tables were very useful but too big
4. Pay attention to the identification of factors (A), (B), (C). They will always have to be the same..... In table 1, (A) and (B) are different from those referred to in tables 2 and 3. You can remove the (A), (B) and (C) and write only the names
5. Beware, it is important to compare the results obtained with the authors presented in
the introduction; in this chapter no author’s are mentioned (for comparison of results),
I advise you to review and introduce some references with similar results.

3. Discussion:
Very good and clear
The discussion is quite elaborate
References are very well introduced in the text and it connects greatly with different authors.

This chapter clarifies (very well) some details not so well clarified in the results, namely in the interpretation of figures and tables.
It presents 18 references that are very connected to the theme and with a sense of opportunity.

4. Materials and Methods:
I don't know the rules of the journal but it seems important that the Material and Methods
chapter be presented before the Results and Discussion to better integrate the reader.
To understand the results I had to read the Material and Methods first, only then did I realize
the location and that the trial had 4 years of data.
· In point 4.2 after identifying the varieties, it should mention the number of years of
trials.
· In point 4.3 (line 338) what does pulses mean? will it be pods?
· And mass of seeds? will it be the weight of the seeds?
In summary the Material and Methods chapter is very clear, containing all the information on
the realization of this work
The methodology used throughout this work is very well presented.

5. Conclusions:
Very clear and well written.
Small but enough

References:
All references are in the text and are very diverse (years and authors).
The work is very strong and rich, as can be seen from the large number of references

GENERAL INFORMATION:
This article is very interesting and very useful for the world, especially for farmers and the scientific community.
It is not a new work because in my institution similar experiments are also carried out.

Author Response

A list of amendments to the Manuscript ID: plants- 2369985

Daiva Janusauskaite

Productivity of three pea (Pisum sativum L.) varieties as influenced by nutrient management

practices and correlation with meteorological conditions in boreal environmental zone

Response to Reviewer 2

Dear colleagues,

we are very appreciative to you for your valuable remarks, advice and your time spared for my manuscript. We made amendments accordingly to your remarks. Please find below our detailed description of the corrections.

Abstract:
Very good and clear
Referred to all topics of the work.
Very well explains the research carried out
It should be mentioned where trials were carried out (experiments) and the number of years
The keywords are very well chosen.

  1. Introduction:
    Good presentation and although not very extensive it is enough to clarifying and understand the rest of the paper.
    Missing square parenthesis on line 44 = [6.13]
    It presents 25 references that are very connected to the theme and with a sense of

     Answer:  We corrected L 44.

  1. Results:
    The results meet the expectations and objectives of the program and are very complete; however, there were some comments:
    They are very long and difficult to interpret and read, but after several readings it
    becomes understandable.
    2. Figure 1 (line 71) does not agree with the text.

   Ans.: We corrected “Figure 1” into “Table 1”.Figure 1 (line 71)

  1. The tables were very useful but too big
    4. Pay attention to the identification of factors (A), (B), (C). They will always have to be the same..... In table 1, (A) and (B) are different from those referred to in tables 2 and 3. You can remove the (A), (B) and (C) and write only the names

Ans.: Two-way ANOVA in Table 1, so there are only two factors. The other tables present a three-way ANOVA, with year as a factor, so there are three factors. Everything is explained at the top of the table, and the letter of the factors in parentheses is necessary because they are also used in the text. Let me not change this.

  1. Beware, it is important to compare the results obtained with the authors presented in
    the introduction; in this chapter no author’s are mentioned (for comparison of results),
    I advise you to review and introduce some references with similar results.

Ans.: Because the sections "Results" and "Discussion" are presented separately, the data of other authors are also compared in the Discussion section, and in the section  Results discusses the data obtained.

  1. Discussion:
    Very good and clear
    The discussion is quite elaborate
    References are very well introduced in the text and it connects greatly with different authors.

This chapter clarifies (very well) some details not so well clarified in the results, namely in the interpretation of figures and tables.
It presents 18 references that are very connected to the theme and with a sense of opportunity.

Ans.: Thank you for the good rating of the "Discussion" section.

  1. Materials and Methods:
    I don't know the rules of the journal but it seems important that the Material and Methods
    chapter be presented before the Results and Discussion to better integrate the reader.
    To understand the results I had to read the Material and Methods first, only then did I realize
    the location and that the trial had 4 years of data.

Ans.: An article template is provided in the requirements for article authors. Arrangement of sections of the article in the template: abstract, introduction, Results, Discussion, Material and Methods, Conclusions. We followed the order in which the parts were laid out in the template.

  • In point 4.2 after identifying the varieties, it should mention the number of years of
    trials.

Ans.: The duration of the experiment is already specified in the line 301: “A field experiment was carried out during 2014-2017 at the Institute of Agriculture, Lithuanian Research Centre for Agriculture and Forestry in Central Lithuania (55°23′50″ N and 23°51′40″ E).“

  • In point 4.3 (line 338) what does pulses mean? will it be pods?

Ans.: We changed ‘pulses’ to ‘pods’ throughout the article.

  • And mass of seeds? will it be the weight of the seeds?

Ans.: We changed “mass” to ‘weight” , ‘MSP’ to ‘WSP’ throughout the article.

In summary the Material and Methods chapter is very clear, containing all the information on
the realization of this work.
The methodology used throughout this work is very well presented.

  1. Conclusions:
    Very clear and well written.
    Small but enough

References:
All references are in the text and are very diverse (years and authors).
The work is very strong and rich, as can be seen from the large number of references

GENERAL INFORMATION:
This article is very interesting and very useful for the world, especially for farmers and the scientific community.
It is not a new work because in my institution similar experiments are also carried out.

Round 2

Reviewer 1 Report

The author has substantially improved the manuscript entitled “Productivity of three pea (Pisum sativum L.) varieties as influenced by nutrient supply and meteorological conditions in boreal environmental zone (plants-2369985). All of my concerns have been addressed. Revisions are satisfactory and I would recommend it for publication in its present form.